# Perplexity-aware Correction for Robust Alignment with Noisy Preferences

**Keyi Kong**[1,*]   **Xilie Xu**[2,*]   **Di Wang**[3]   **Jingfeng Zhang**[4,5,†]   **Mohan Kankanhalli**[2]

[1]Shandong University  [2]National University of Singapore
[3]King Abdullah University of Science and Technology  [4]The University of Auckland
[5]RIKEN Center for Advanced Intelligence Project (AIP)

`luxinyayaya@mail.sdu.edu.cn, xuxilie@comp.nus.edu.sg`
`di.wang@kaust.edu.sa, jingfeng.zhang@auckland.ac.nz, mohan@comp.nus.edu.sg`

## Abstract

Alignment techniques are critical in ensuring that large language models (LLMs) output helpful and harmless content by enforcing the LLM-generated content to align with human preferences. However, the existence of noisy preferences (NPs), where the responses are mistakenly labelled as chosen or rejected, could spoil the alignment, thus making the LLMs generate useless and even malicious content. Existing methods mitigate the issue of NPs from the loss perspective by adjusting the alignment loss based on a clean validation dataset. Orthogonal to these loss-oriented methods, we propose perplexity-aware correction (PerpCorrect) from the data perspective for robust alignment which detects and corrects NPs based on the differences between the perplexity of the chosen and rejected responses (dubbed as PPLDiff). Intuitively, a higher PPLDiff indicates a higher probability of the NP because a rejected/chosen response which is mistakenly labelled as chosen/rejected is less preferable to be generated by an aligned LLM, thus having a higher/lower perplexity. PerpCorrect works in three steps: (1) PerpCorrect aligns a surrogate LLM using the clean validation data to make the PPLDiff able to distinguish clean preferences (CPs) and NPs. (2) PerpCorrect further aligns the surrogate LLM by incorporating the reliable clean training data whose PPLDiff is extremely small and reliable noisy training data whose PPLDiff is extremely large after correction to boost the discriminatory power. (3) Detecting and correcting NPs according to the PPLDiff obtained by the aligned surrogate LLM to obtain a denoised training dataset for robust alignment. Comprehensive experiments validate that our proposed PerpCorrect can achieve state-of-the-art alignment performance under NPs. Notably, PerpCorrect demonstrates practical utility by requiring only a modest amount of validation data and being compatible with various alignment techniques. Our code is available at PerpCorrect.

## 1 Introduction

Alignment enables the safe utilization of the remarkable capabilities acquired by large language models (LLMs) through self-supervised learning on vast corpora [6, 24, 4]. It refers to the process of ensuring that the contents generated by LLMs are helpful, harmless, and aligned with human values and preferences [19]. Reinforcement Learning from Human Feedback (RLHF) [9] has emerged as a primary technique for achieving alignment. Current technical routes [39, 40, 29] require a reward model to simulate human preference and use it to optimize the policy model outputs with

---

[*] Equal contribution.
[†] Corresponding author.

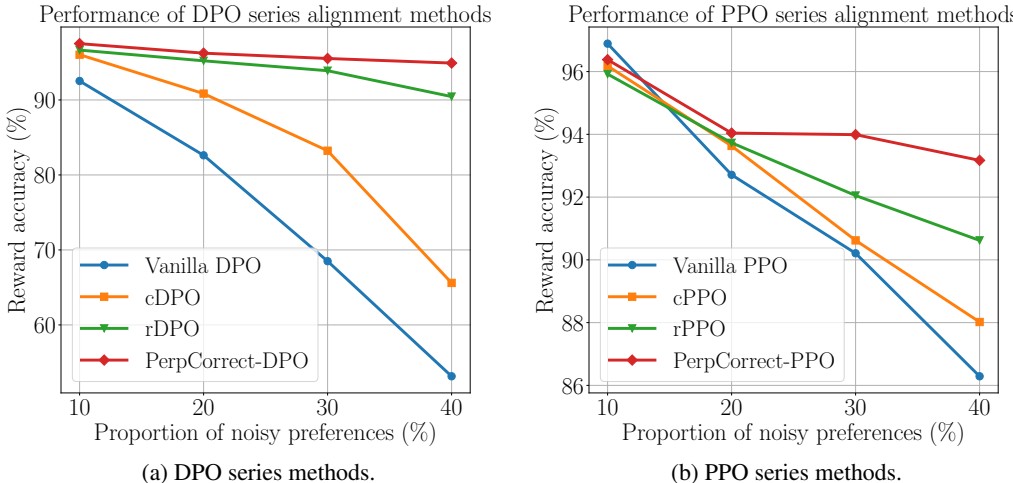

(a) DPO series methods.       (b) PPO series methods.

Figure 1: We evaluated various robust alignment methods under different proportions of noisy preferences using the Llama2-7B model, on the Golden HH dataset. The reward accuracy of both the vanilla DPO and PPO method significantly decreases as the proportion of noisy preferences increases. Our method, perplexity-aware correction (PerpCorrect), outperforms both the DPO and PPO series baselines across different proportions of noisy preferences.

Proximal Policy Optimization (PPO) [27]. Current offline techniques such as Direct Preference Optimisation (DPO) [26], Sequence Likelihood Calibration with Human Feedback (SLiC) [38] and Identity-Preference Optimisation (IPO) [3], could directly align LLMs without intensively training a reward model as employed in RLHF.

Recent studies [34, 8] have shown there exist noisy preferences (NPs) that may lead to significant degradation in alignment performance. The issue of NPs, where the label of the actually chosen/rejected responses in training datasets is flipped as rejected/chosen, can arise from the biases of annotators [34] and malicious noise injection [5]. As shown in Figure 1, when NPs are randomly injected into the training dataset, the conventional alignment method (e.g., DPO [26] and PPO [9]) will have significantly degraded alignment performance measured by the reward accuracy. Such performance degradation could result in the generation of useless and even malicious content [34]. Therefore, it necessitates developing robust alignment methods that can utilize datasets with NPs to effectively align the LLMs with human preferences.

Existing robust alignment methods are proposed from the loss perspective, which adjust the alignment loss using a clean validation dataset to mitigate the issue of NPs. In particular, the conservative DPO (cDPO) [21] and robust-DPO (rDPO) [8] both estimate the proportion of NPs using the clean validation data via cross-validation and then adjust the original DPO loss based on the estimated proportion of NPs. However, Mitchell [21] and Chowdhury et al. [8] overlooked the essential differences between noisy and clean preferences, which is critical for mitigating the issue of NPs.

To this end, we propose **Perp**lexity-aware **Correct**ion (PerpCorrect) for robust alignment from the data perspective by leveraging the differences between noisy and clean preferences for robust alignment. PerpCorrect detects and corrects NPs based on the difference between the perplexity of the chosen response and that of the rejected counterparts (dubbed as PPLDiff) obtained by an aligned surrogate LLM using the clean validation set. If an NP is detected, PerpCorrect will correct it by flipping the label of the rejected/chosen responses as chosen/rejected. Intuitively, rejected responses which are mistakenly labelled as chosen have a higher perplexity since they are less consistent with human preferences and thus have a lower probability of being generated after alignment. Therefore, a higher value of PPLDiff indicates a higher probability of the preferences being noisy. In this way, PerpCorrect leverages the differences between noisy and clean preferences (CPs) identified by PPLDiff to detect NPs.

To make the PPLDiff able to distinguish CPs and NPs, PerpCorrect requires an aligned surrogate LLM for calculating PPLDiff. The density of PPLDiff obtained on the noisy training dataset using

an unaligned surrogate LLM, which can be fitted as a normal distribution centered around zero (evidenced in Figure 2a), cannot discriminate CPs and NPs. Therefore, we align a surrogate LLM using the clean validation data. The density of PPLDiff obtained by the aligned surrogate LLM in Figure 2b can be fitted into two distinguishable normal distributions, thus being able to differentiate CPs and NPs.

However, there still exists a large overlap between two normal distributions after aligning only on the clean validation dataset, which could result in an unsatisfactory accuracy of NP detection. To this end, we iteratively align the model using more reliable clean training data with extremely low PPLDiff (located in the green area in Figure 2c) and reliable noisy training data with extremely large PPLDiff (located in the red area in Figure 2c) sampled from noisy training datasets. Finally, the two normal distributions are significantly separated as shown in Figure 2d, which indicates that PPLDiff has an enhanced discriminatory power.

Benefiting from the strong discriminatory power of PPLDiff calculated by the aligned surrogate LLM, PerpCorrect outputs a denoised training dataset for robust alignment by first detecting NPs based on a PPLDiff threshold and then correcting them. The data, whose PPLDiff is below a certain threshold (i.e., the black dotted line in Figure 2d) selected as the x-coordinate of the two normal distributions' intersection, are identified as NPs and thus corrected by flipping the response's label. Notably, our proposed PerpCorrect is compatible with various alignment methods as well as robust alignment methods [21, 8] since the metric PPLDiff is agnostic to training algorithms and only requires an arguably small number of clean validation data (~50), thus making it practical.

Comprehensive empirical results, evaluated using the Llama2-7B [32] and phi-2 [20] models on the OpenAssistant Conversations (OASST1) [17] and Golden HH [7] datasets, validate the effectiveness of our proposed PerpCorrect method in robustifying alignment with NPs. We empirically validate that PerpCorrect consistently obtains state-of-the-art performance among various proportions of NPs. Besides, we empirically demonstrate that PerpCorrect can effectively robustify various alignment techniques and robust alignment methods, validating its compatibility.

## 2 Literature Review and Preliminary

In this section, we introduce the related work about LLM alignment and provide preliminaries about the noisy preferences, perplexity, as well as various alignment methods.

### 2.1 LLM Alignment

In the domain of aligning LLMs with human preferences, pairwise preference methods are favored due to their lower cognitive burden on evaluators. Traditional online alignment approaches [32, 24, 29] involve training reward models from these preferences to provide signals in reinforcement learning. Recent offline alignment methods like Direct Preference Optimization (DPO) [26], Sequence Likelihood Calibration (SLiC) [38], and Identify Preference Optimization (IPO) [3] streamlined this process by directly using preference pairs to train LLMs, thus enhancing performance and reducing computational costs. Additionally, methods like RRHF [37] align LLMs using multiple ranked preferences, Kahneman-Tversky Optimization (KTO) [13] align LLMs using a single preference labeled as good or bad, and Rejection Sampling Optimization (RSO) [18] address DPO's limitation in sampling preference pairs from the optimal policy through rejection sampling. However, NPs, arising from the biased human feedback, can determine the alignment performance [24, 34]. Robust alignment methods like conservative DPO (cDPO) [21], robust DPO (rDPO) [8] have been proposed to address these issues from the loss perspective. Our approach focuses on the data perspective to address these issues of NPs and is orthogonal to these robust alignment methods.

### 2.2 Preliminary

**Noisy preferences (NPs).** NPs refer to preference data in training datasets, whose label of the actually chosen/rejected responses is flipped as rejected/chosen. Let $\mathcal{D} = \{(x^{(i)}, y_w^{(i)}, y_l^{(i)})\}_{i=1}^N$ be the preference dataset consisting of $N \in \mathbb{N}$ preference data points. For each preference data point $(x, y_w, y_l) \in \mathcal{D}$, $x$ is the prompt input to LLMs, $y_w$ is the chosen response, and $y_l$ is the rejected response. We let $\tilde{\mathcal{D}} = \{(x^{(i)}, \tilde{y}_w^{(i)}, \tilde{y}_l^{(i)})\}_{i=1}^N$ be the noisy preference dataset (i.e., preference dataset

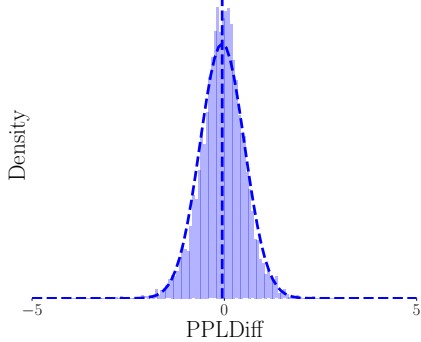
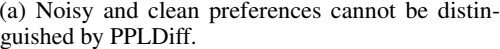
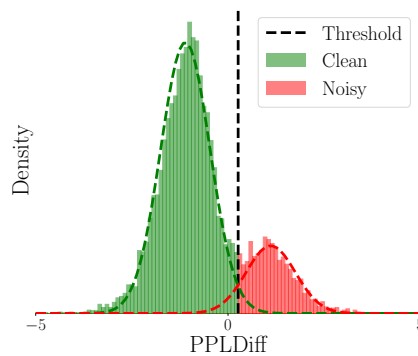

(a) Noisy and clean preferences cannot be distinguished by PPLDiff.

(b) The large overlap between two distributions leads to flawed NP detection.

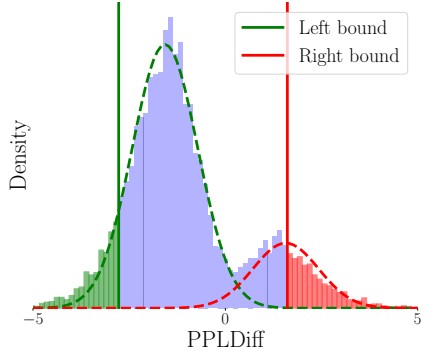
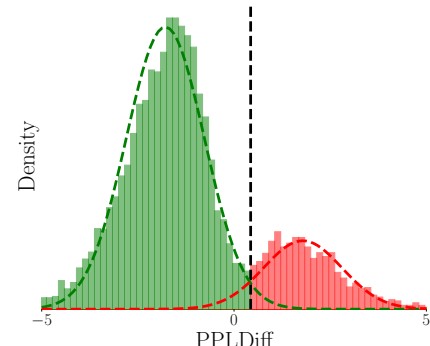

(c) Aligning the surrogate LLM using extra reliable training data.

(d) Separating and correcting noisy preferences based on the threshold.

Figure 2: We visualized the PPLDiff under the entire PerpCorrect process using Llama2-7B on Golden HH dataset with 20% noisy preferences. We use the green dotted line to represent the normal distribution formed by clean data, the red dotted line represents the normal distribution formed by noisy data, and the black dotted line represents the threshold.

consisting noisy preferences) and denote preference data points that are not noisy as clean preferences (CPs). Following Chowdhury et al. [8], we obtain the noisy preference dataset $\tilde{\mathcal{D}}$ using the standard random noise model [23] with the probability $\varepsilon \in (0, 50\%)$ to change the data point into noisy preferences, i.e.

$$\mathbb{P}_{(x^{(i)}, \tilde{y}_w^{(i)}, \tilde{y}_l^{(i)}) \sim \tilde{\mathcal{D}}} \left[ (x^{(i)}, \tilde{y}_w^{(i)}, \tilde{y}_l^{(i)}) = (x^{(i)}, y_l^{(i)}, y_w^{(i)}) \right] = \varepsilon. \tag{1}$$

**Perplexity (PPL).**    PPL [15] measures the probability that the LLM generates a sentence. A lower PPL of a sentence indicates that the LLM has generated this sentence with a high probability. PPL is defined as the average negative log-likelihood of a sequence, i.e.,

$$\mathrm{PPL}(s; \theta) = \exp(-\frac{1}{t} \sum_{i=1}^{t} \log \pi_\theta(s_i | s_{<i})), \tag{2}$$

where $s$ is a sequence composed of $t$ tokens and $\log \pi_\theta(s_i | s_{<i})$ denotes the log-likelihood of the $i$-th token given the preceding tokens $s_{<i}$ calculated by an LLM $\pi_\theta$.

**Technical details of alignment methods.**    There are usually three phases in RLHF pipeline [34, 26]: (1) supervised fine-tuning (SFT); (2) reward modeling; (3) reinforcement learning (RL) optimization. In the SFT phase, an LLM is fine-tuned via supervised learning on high-quality task-related data.

We denote the LLM after the SFT phase as $\pi_{\text{SFT}}$. In the reward modeling phase, the reward model is introduced to simulate human preferences. Given a preference dataset, a reward model $r_\omega(x, y)$ parameterized by $\omega$, which takes prompt $x$ and response $y$ as input and outputs a real number representing the reward score, can be optimized via minimizing the following loss function:

$$\mathcal{L}_R(r_\omega, \mathcal{D}) = -\mathbb{E}_{(x, y_w, y_l) \sim \mathcal{D}} \left[ \log \sigma(r_\omega(x, y_w) - r_\omega(x, y_l)) \right], \tag{3}$$

where $\sigma$ is the logistic function. In the RL optimization phase, the objective function is as follows:

$$\max_\theta \mathbb{E}_{x \sim \mathcal{D}, y \sim \pi_\theta(y|x)} [r_\omega(x, y) - \beta \cdot (\log \pi_\theta(y|x) - \log \pi_{\text{ref}}(y|x))], \tag{4}$$

where $\pi_\theta(y|x)$ represents the probability that the LLM parameterized by $\theta > 0$ generates the response $y$ given the prompt $x$, $\pi_{\text{ref}}$ is a reference LLM to maintain the generation ability of the aligned model, and $\beta$ is a hyper-parameter to ensure the similarity between $\pi_\theta(y \mid x)$ and $\pi_{\text{ref}}(y \mid x)$. We take $\pi_{\text{SFT}}$ as the reference LLM $\pi_{\text{ref}}$ following Ouyang et al. [24].

Recently, offline alignment methods directly leverages preferences in preference datasets, bypassing the need to learn a reward model in RLHF. The LLM parameters are optimized by minimizing the following loss function:

$$\mathcal{L}(\pi_\theta; \pi_{\text{ref}}) = \mathbb{E}_{(x, y_w, y_l) \sim \mathcal{D}} \left[ \mathcal{G}(x, y_w, y_l; \theta) \right], \tag{5}$$

where the function $\mathcal{G}$ changes with the alignment method. To be specific, DPO [26] uses a BCE loss, SLiC [38] uses a hinge loss, and IPO [3] uses a square loss:

$$\mathcal{G}_{\text{DPO}}(x, y_w, y_l; \theta) = -\log \sigma \left( \beta \log \frac{\pi_\theta(y_w \mid x)}{\pi_{\text{ref}}(y_w \mid x)} - \beta \log \frac{\pi_\theta(y_l \mid x)}{\pi_{\text{ref}}(y_l \mid x)} \right), \tag{6}$$

$$\mathcal{G}_{\text{SLiC}}(x, y_w, y_l; \theta) = \max \left\{ 0, 1 - \left( \beta \log \frac{\pi_\theta(y_w \mid x)}{\pi_{\text{ref}}(y_w \mid x)} - \beta \log \frac{\pi_\theta(y_l \mid x)}{\pi_{\text{ref}}(y_l \mid x)} \right) \right\}, \tag{7}$$

$$\mathcal{G}_{\text{IPO}}(x, y_w, y_l; \theta) = \left( \beta \log \frac{\pi_\theta(y_w \mid x)}{\pi_{\text{ref}}(y_w \mid x)} - \beta \log \frac{\pi_\theta(y_l \mid x)}{\pi_{\text{ref}}(y_l \mid x)} - \frac{1}{2} \right)^2. \tag{8}$$

To mitigate the issue of NPs, cDPO [21] and rDPO [8] adjust the DPO loss based on the estimated proportion of NPs $\varepsilon'$ using a clean validation dataset $\mathcal{D}_{\text{val}} = \{(x^{(i)}, y_w^{(i)}, y_l^{(i)})\}_{i=1}^{N_{\text{val}}}$ consisting of $N_{\text{val}} \in \mathcal{N}$ clean preference data points, i.e.

$$\mathcal{G}_{\text{cDPO}}(x, \tilde{y}_w, \tilde{y}_l; \theta) = (1 - \varepsilon')\mathcal{G}_{\text{DPO}}(x, \tilde{y}_w, \tilde{y}_l; \theta) + \varepsilon'\mathcal{G}_{\text{DPO}}(x, \tilde{y}_l, \tilde{y}_w; \theta), \tag{9}$$

$$\mathcal{G}_{\text{rDPO}}(x, \tilde{y}_w, \tilde{y}_l; \theta) = \frac{(1 - \varepsilon')\mathcal{G}_{\text{DPO}}(x, \tilde{y}_w, \tilde{y}_l; \theta) - \varepsilon'\mathcal{G}_{\text{DPO}}(x, \tilde{y}_l, \tilde{y}_w; \theta)}{1 - 2\varepsilon'}. \tag{10}$$

## 3 Perplexity-aware Correction for Robust Alignment

This section introduces **Perp**lexity-aware **Correct**ion (PerpCorrect) for robust alignment with NPs. In Section 3.1, we introduce a novel metric called PPLDiff and then illustrate the pipeline of PerpCorrect to detect and correct NPs based on PPLDiff. In Section 3.2, we demonstrate how to adapt our proposed PerpCorrect with various alignment methods to achieve robust alignment.

### 3.1 Perplexity-aware Correction (PerpCorrect)

In this subsection, we introduce PerpCorrect which employs a novel metric called PPLDiff as the foundation for detecting and correcting NPs. The algorithm of PerpCorrect is described in Algorithm 2.

**PPLDiff.** PPLDiff measures the difference between the PPL of chosen response and that of the rejected response. Given a preference data point $(x, \tilde{y}_w, \tilde{y}_l) \in \tilde{\mathcal{D}}$ sampled from the noisy training dataset $\tilde{\mathcal{D}}$ and an LLM $\pi_\theta$, PPLDiff is defined as follows:

$$\text{PPLDiff}(x, \tilde{y}_w, \tilde{y}_l; \theta) = \log \text{PPL}([x; \tilde{y}_w]; \theta) - \log \text{PPL}([x; \tilde{y}_l]; \theta), \tag{11}$$

where $[x; y]$ indicates the concatenation of the prompt $x$ and the response $y$. Intuitively, if a data point is a clean preference, the $\text{PPL}([x; \tilde{y}_w]; \theta)$ will be lower than $\text{PPL}([x; \tilde{y}_l]; \theta)$ because the sequence $[x; \tilde{y}_w]$ is more aligned with human values and thus has a higher probability of being generated by aligned LLMs. As a result, it PPLDiff will be lower compared to NPs, which $\text{PPL}([x; \tilde{y}_w]; \theta)$ is higher than $\text{PPL}([x; \tilde{y}_l]; \theta)$. This difference allows us to distinguish CPs and NPs based on PPLDiff.

**Aligning a surrogate LLM only using clean validation data.** Here, we utilize a clean validation dataset $\mathcal{D}_{\mathrm{val}}$ to obtain an aligned surrogate LLM to make PPLDiff able to distinguish CPs and NPs. We empirically find that the PPLDiff values of CPs and NPs calculated by an unaligned LLM in the noisy training dataset were initially indistinguishable as shown in Figure 2a, making it impossible to differentiate the NPs from CPs. This is because an unaligned LLM lacks the necessary preferences to distinguish NPs and CPs.

Therefore, we introduce a surrogate LLM $\pi_{\theta'}$ parameterized by $\theta'$ to replace the unaligned LLM and use it for calculating PPLDiff. We optimize the surrogate LLM $\pi_{\theta'}$ using the clean validation dataset $\mathcal{D}_{\mathrm{val}}$ as follows:

$$\max_{\theta'} \mathbb{E}_{(x,y_w,y_l)\sim\mathcal{D}_{\mathrm{val}}} \left[ \mathcal{G}_{\mathrm{DPO}}(x, y_w, y_l; \theta') \right]. \tag{12}$$

After aligning the surrogate LLM, the PPLDiff values of NPs calculated by the surrogate LLM $\pi_{\theta'}$ are significantly increased and those of CPs are significantly decreased, forming two distinct distributions as shown in Figure 2b. This is because the aligned surrogate LLM is trained to generate responses that align with human preferences, enhancing its ability to distinguish between NPs and CPs based on PPLDiff.

To separate CPs and NPs in the noisy training dataset without knowing the oracle preferences, we leverage the Levenberg-Marquardt (LM) algorithm to find two normal distributions that fit the density of PPLDiff calculated by the aligned surrogate LLM. Specifically, the LM algorithm returns the constants $\bar{\varepsilon}, \bar{\mu}, \bar{\sigma}$ that satisfies the following condition:

$$h(x|\bar{\varepsilon}, \bar{\mu}, \bar{\sigma}) = (1 - \bar{\varepsilon}) f_{\mathrm{clean}}(x|\bar{\mu}, \bar{\sigma}^2) + \varepsilon f_{\mathrm{noisy}}(x| - \bar{\mu}, \bar{\sigma}^2), \tag{13}$$

$$\text{where} \quad f(x|\mu, \sigma^2) = \frac{1}{\sqrt{2\sigma^2\pi}} \exp(-\frac{(x-\mu)^2}{2\sigma^2}). \tag{14}$$

Note that $x$ is the PPLDiff value and $h(x|\bar{\varepsilon}, \bar{\mu}, \bar{\sigma})$ is the superposition of these two normal distribution. We denote $f_{\mathrm{clean}}(x|\bar{\mu}, \bar{\sigma}^2)$ as the normal distribution fitting the PPLDiff of CPs and $f_{\mathrm{noisy}}(x| - \bar{\mu}, \bar{\sigma}^2)$ as the normal distribution fitting the PPLDiff of NPs since the PPLDiff of NPs is intuitively higher than that of CPs. In this way, we can obtain two distinguishable normal distributions to separate NPs and CPs as shown in the green and red dotted lines of Figure 2b without knowing the oracle preferences.

**Further aligning the surrogate LLM using extra reliable training data from noisy training datasets.** After aligning only using the clean validation datasets, the discriminatory power of the PPLDiff is still far from satisfactory because of the large overlap between the two normal distributions. Therefore, we align the surrogate LLM with more reliable training data to make the PPLDiff of CPs and that of NPs more separable. We iteratively align the surrogate LLM $\pi_{\theta'}$ using more reliably clean training data whose PPLDiff is extremely small and reliably noisy training data whose PPLDiff is extremely large after correction by flipping the label of the response.

Specifically, at epoch $t \in \mathbb{N}$, we select $(t - 1) \cdot \alpha \cdot |\tilde{\mathcal{D}}|$ of the training data along with the clean validation data for further alignment where $\alpha \in (0, 1)$ is the selection ratio and $|\tilde{\mathcal{D}}| = N$ is the number of data points in noisy training dataset. As shown in Lines 33–45 of Algorithm 2, the selected reliable training dataset $\mathcal{D}'_t$ consists of $(t - 1) \cdot \alpha \cdot (1 - \bar{\varepsilon}) \cdot |\tilde{\mathcal{D}}|$ reliably clean training data whose PPLDiff values are smallest $(t - 1) \cdot \alpha \cdot (1 - \bar{\varepsilon})$ percent and $(t - 1) \cdot \alpha \cdot \bar{\varepsilon} \cdot |\tilde{\mathcal{D}}|$ reliably noisy training data after correction. Note that the reliably clean training data are the data points whose PPLDiff values are smallest $(t - 1) \cdot \alpha \cdot (1 - \bar{\varepsilon})$ percent (located in the green area of Figure 2c), and the reliably noisy training data whose PPLDiff values are largest $(t - 1) \cdot \alpha \cdot \bar{\varepsilon}$ percent (located in the red area of Figure 2c) among all the training data points.

**Detecting and correcting NPs based on PPLDiff to output a denoised training dataset.** Based on the PPLDiff calculated by the aligned surrogate LLM, PerpCorrect detects and corrects NPs whose PPLDiff value is lower than a certain threshold. We take the x-coordinate of the intersection of the two normal distributions as the threshold (the black dotted line in Figure 2d). As shown in Lines 23–31, data points whose PPLDiff values are larger than this threshold are identified as CPs (the green area in Figure 2d), and other data points are identified as NPs requiring correction (the red area in Figure 2d). In this way, we can obtain a denoised training dataset for robust alignment.

---

**Algorithm 1** Robust Alignment via Perplexity-aware Correction (PerpCorrect)

---

1: **Input:** Noisy training dataset $\tilde{\mathcal{D}}$, clean validation dataset $\mathcal{D}_{\text{val}}$, and pre-trained LLM $\pi_\theta$ parameterized by $\theta$
2: **Output:** Robust alignment model $\pi_\theta$
3: // Stage I: Supervised fine-tuning (SFT)
4: $\pi_\theta \leftarrow$ Supervised fine-tuned LLM $\pi_\theta$. (Details in Appendix C.3)
5: // Stage II: Perplexity-aware correction using the surrogate LLM
6: $\tilde{\mathcal{D}}_{\text{denoised}}, \varepsilon'_{\text{denoised}} \leftarrow$ Perplexity-aware Correction ($\pi_\theta, \tilde{\mathcal{D}}, \mathcal{D}_{\text{val}}$) (Details in Algorithm 2)
7: // Stage III: Alignment with denoised dataset
8: $\pi_\theta \leftarrow$ Aligned LLM $\pi_\theta$ using $\tilde{\mathcal{D}}_{\text{denoised}}$ and $\varepsilon'_{\text{denoised}}$ (Details in Appendix C.3)

---

Further, we select an optimal denoised training dataset to further enhance the performance of robust alignment according to the intersection area of the two normal distributions. We denote the intersection area of two normal distributions as the estimated NP proportion of the denoised training dataset, i.e.,

$$\varepsilon'_{PC} = \int_{-\inf}^{+\inf} \min\{(1-\bar{\varepsilon})f_{\text{clean}}(x|\bar{\mu}, \bar{\sigma}^2), \bar{\varepsilon}f_{\text{noisy}}(x|-\bar{\mu}, \bar{\sigma}^2)\}\mathrm{d}x, \tag{15}$$

where $\varepsilon'_{PC}$ calculates the ratio of noisy data points which are not detected by PerpCorrect (i.e., the green area enclosed by the black and red lines in Figure 2d) and the clean data points which are mistakenly detected by PerpCorrect (i.e., the red area enclosed by the black and green lines in Figure 2d). In this way, $\varepsilon'_{PC}$ can efficiently calculate the NP proportion of the denoised training dataset. We take the denoised training dataset with the smallest $\varepsilon'_{PC}$ among multiple iterations as the optimal one for robust alignment to boost alignment performance.

## 3.2 Robust Alignment

Here, we introduce how to adapt PerpCorrect to robustify various alignment methods and demonstrate the algorithm of robust alignment via PerpCorrect in Algorithm 1. In general, the pipeline of the robust alignment based on PerpCorrect contains three stages: SFT, PerpCorrect, and alignment. We will first conduct SFT, following Christiano et al. [9], to boost the performance of a pre-trained LLM by boosting its skills for specific tasks. Next, we will conduct PerpCorrect to detect and correct NPs and output an optimal denoised training dataset $\tilde{\mathcal{D}}_{\text{denoised}}$ the smallest $\varepsilon'_{PC}$ in Eq. 15. Finally, we can obtain an aligned LLM from the SFT model using the denoised training dataset $\tilde{\mathcal{D}}_{\text{denoised}}$ via alignment (i.e., Line 8 in Algorithm 1).

Because our proposed PerpCorrect is agnostic to alignment methods and model structures, PerpCorrect is applicable to robustify both online alignment methods such as RLHF (PPO) [9] and offline alignment methods including DPO [26], SLiC [38], and IPO [3]. Besides, our proposed PerpCorrect is compatible with existing loss-oriented robust alignment methods, such as cDPO [21] and rDPO [8], based on the estimated proportion of NPs. Note that cDPO and rDPO require conducting computationally expensive cross-validation to tune the estimated proportion of NPs. We can efficiently estimate the proportion of NPs by utilizing the fitted normal distributions during PerpCorrect, i.e., $\varepsilon'_{PC}$ in Eq. 15. Therefore, we can combine PerpCorrect with a wide range of existing alignment methods to achieve robust alignment with NPs.

# 4 Experiments

In this section, we demonstrate that our proposed PerpCorrect achieves state-of-the-art alignment performance under different proportion of NPs and have good compatibility with other alignment methods. In Section 4.1, PerpCorrect combined with DPO [26] achieves state-of-the-art alignment performance than existing baselines (Section 4.1), including DPO [26], cDPO [21], and rDPO [8]. In Section 4.2, we further analyze the impact of the number of validation data and verified the compatibility of PerpCorrect with online and offline alignment methods and robust alignment methods. The training details and compute resources are reported in Appendix C.1.

Table 1: Average reward accuracy of DPO series alignment methods using Llama2-7B on the Golden HH dataset. The standard deviation of reward accuracy is reported in Table 8.

| Method | Proportion of noisy preferences (%) | | | |
|---|---|---|---|---|
| | 10 | 20 | 30 | 40 |
| Vanilla DPO | 92.53% | 82.62% | 68.50% | 53.15% |
| cDPO | 96.04% | 90.85% | 83.23% | 65.60% |
| rDPO | 96.65% | 95.22% | 93.90% | 90.45% |
| PerpCorrect-DPO | **97.51%** | **96.24%** | **95.53%** | **94.92%** |

Table 2: Average reward accuracy of PPO series alignment methods using Llama2-7B on the Golden HH dataset. The standard deviation of reward accuracy is reported in Table 9.

| Method | Proportion of noisy preferences (%) | | | |
|---|---|---|---|---|
| | 10 | 20 | 30 | 40 |
| Vanilla PPO | **96.64%** | 92.71% | 90.21% | 86.29% |
| cPPO | 96.18% | 93.63% | 90.62% | 88.02% |
| rPPO | 95.92% | 93.73% | 92.05% | 90.62% |
| PerpCorrect-PPO | 96.38% | **94.04%** | **93.99%** | **93.17%** |

Table 3: Average reward accuracy of DPO series alignment methods using phi-2 on the Golden HH dataset. The standard deviation of reward accuracy is reported in Table 10.

| Method | Proportion of noisy preferences (%) | | | |
|---|---|---|---|---|
| | 10 | 20 | 30 | 40 |
| Vanilla DPO | 93.19% | 85.57% | 73.07% | 54.98% |
| cDPO | 97.21% | 92.63% | 81.05% | 66.72% |
| rDPO | 96.49% | 95.73% | 93.34% | 84.55% |
| PerpCorrect-DPO | **98.17%** | **97.05%** | **97.66%** | **96.39%** |

Table 4: Average reward accuracy of DPO series alignment methods using phi-2 on the OASST1 dataset. The standard deviation of reward accuracy is reported in Table 11.

| Method | Proportion of noisy preferences (%) | | | |
|---|---|---|---|---|
| | 10 | 20 | 30 | 40 |
| Vanilla DPO | 66.94% | 62.61% | 58.44% | 52.42% |
| cDPO | 67.30% | 61.44% | 54.87% | 49.21% |
| rDPO | 63.95% | 59.47% | 56.45% | 45.20% |
| PerpCorrect-DPO | **71.34%** | **69.04%** | **68.27%** | **68.49%** |

**Datasets.** We utilize two preference datasets, namely OpenAssistant Conversations (OASST1) [17] and Golden HH [7]. The processed OASST1 dataset comprises 17,939 training samples and 951 testing samples and the processed Golden HH dataset consists of 12,066 training samples and 654 testing samples. The description and processing details of these datasets are provided in Appendix C.2.

**Models.** Our evaluation leverages two distinct series of open-sourced LLMs with different parameter sizes: Llama2-7B [32] and phi-2 [20]. We acquire the checkpoints from their official repositories on Hugging Face. The LLMs used for PerpCorrect and those for robust alignment share the same model structure and initialization.

**Baselines.** We adopt vanilla DPO [26] and two robust alignment methods, cDPO [21] and rDPO [8], as baselines. For their detailed implementation, we utilize and adapt the transformers and TRL libraries provided by the Hugging Face community.

**Metrics.** In accordance with Chowdhury et al. [8], we employ the winning rate of policy generations against the selected preferences on the test dataset as our primary metric. This metric applies to vanilla DPO [26], cDPO [21], rDPO [8], as well as other offline alignment methods including SLiC [38] and IPO [3]. Additionally, we utilize the winning rate of the reward model score for the chosen preferences on the test dataset as our metric for vanilla PPO [24], cPPO [21, 34], and rPPO [8]. These two metrics are collectively called reward accuracy.

### 4.1 PerpCorrect Achieves the State-of-the-Art Robust Alignment Performance

The empirical results demonstrate that our method, PerpCorrect, achieves state-of-the-art robust alignment performance, surpassing existing baselines such as vanilla DPO [26], cDPO [21], and rDPO [8]. This is evident across various proportions of noisy preferences $\varepsilon$ using different datasets and LLMs.

**Comparison using different LLMs.** Tables 1 and 3 demonstrate the average reward accuracy of the DPO series alignment methods on the Golden HH [7] dataset using Llama2-7B [32] and phi-2 [20]. At a proportion of the NPs $\varepsilon = 40\%$, PerpCorrect increases the reward accuracy by 41.77% (from 53.15% to 94.92%) using Llama2-7B and by 41.41% (from 54.98% to 96.39%) using phi-2. The empirical result validates that our proposed PerpCorrect can be used on different sizes of LLMs and achieve better alignment performance than baselines.

**Comparison on different datasets.** Tables 3 and 4 demonstrate the average reward accuracy of the DPO series alignment methods on the Golden HH [7] and OASST1 [17] datasets using phi-2 [20].

Table 5: Impact of the number of clean validation data evaluated on the Golden HH dataset using Llama2-7B with a proportion of NPs $\varepsilon = 40\%$.

| Number | 10 | 20 | 30 | 40 | 50 | 100 | 200 |
|---|---|---|---|---|---|---|---|
| Reward accuracy | 81.40% | 88.26% | 94.21% | 94.21% | 95.43% | 95.43% | 96.04% |

Table 6: Average reward accuracy and improvements of the offline and robust alignment methods, as well as those combined with PerpCorrect, using Llama2-7B on the Golden HH dataset. The standard deviation of reward accuracy and improvements is reported in Table 12.

| Method | Proportion of noisy preferences (%) | | | |
|---|---|---|---|---|
| | 10 | 20 | 30 | 40 |
| DPO | 92.53% | 82.62% | 68.50% | 53.15% |
| PerpCorrect-DPO | 97.51% | 96.24% | 95.53% | 94.92% |
| $\Delta$ | **+4.98%** | **+13.62%** | **+27.03%** | **+41.77%** |
| SLiC | 96.70% | 87.75% | 76.17% | 58.59% |
| PerpCorrect-SLiC | 96.95% | 95.02% | 95.38% | 94.61% |
| $\Delta$ | **+0.25%** | **+7.27%** | **+19.21%** | **+36.02%** |
| IPO | 98.07% | 92.73% | 79.17% | 61.64% |
| PerpCorrect-IPO | **98.73%** | **97.66%** | **97.82%** | **97.56%** |
| $\Delta$ | **+0.66%** | **+4.93%** | **+18.65%** | **+35.92%** |
| cDPO | 96.04% | 90.85% | 83.23% | 65.60% |
| PerpCorrect-cDPO | 98.12% | 97.31% | 94.97% | 88.36% |
| $\Delta$ | **+2.08%** | **+6.46%** | **+11.74%** | **+22.76%** |
| rDPO | 96.65% | 95.22% | 93.90% | 90.45% |
| PerpCorrect-rDPO | 95.99% | 95.02% | 94.77% | 95.73% |
| $\Delta$ | -0.66% | -0.20% | **+0.87%** | **+5.28%** |

The empirical results reveal a significant discrepancy in average reward accuracy between the more complex OASST1 dataset and the Golden HH dataset. The performance of other robust alignment methods is found to be unsatisfactory on the OASST1 dataset, often not surpassing the vanilla DPO. In contrast, our method PerpCorrect consistently maintains strong alignment performance across varying proportions of noisy preferences. In general, our method PerpCorrect can achieve better alignment performance than baselines across different datasets.

## 4.2 Ablation Study

**Impact of the number of clean validation data.** Table 5 illustrates the impact of the number of clean validation data points. We conducted experiments on the Golden HH dataset using Llama2-7B with a proportion of NPs $\varepsilon = 40\%$. The empirical results indicate that as the number of clean validation data points increases, the performance of our method, PerpCorrect, also improves. However, when the number is too large, the improvement in performance is not obvious, and the cost of manual annotation significantly increases.

**Compatibility with online alignment method RLHF (PPO).** We adopt vanilla PPO [24], cPPO [21, 34], and rPPO [8] as baselines. Table 2 shows the alignment performance of PPO series alignment methods on the Golden HH [7] dataset using Llama2-7B. Although vanilla PPO has good performance when the proportion of NPs is low, it still declines significantly when the proportion is high. PerpCorrect maintains desirable alignment performances when the proportion of NPs is high. Our empirical results show that PerpCorrect has desirable compatibility with online alignment method RLHF (PPO).

**Compatibility with various offline alignmet methods.** Table 6 presents the average reward accuracy and improvements of original offline alignment methods compared to those combined with PerpCorrect. Our experiments, conducted on the Golden HH dataset using Llama2-7B, reveal that the reward accuracy of SLiC [38] and IPO [3] both significantly decrease as the proportion of NPs increases, similar to vanilla DPO [26]. However, our method PerpCorrect enhances their alignment performance across different proportions of NPs. Notably, IPO combined with PerpCorrect achieves

the best alignment performance. We conjecture the main reason is that the proportion of NPs in the denoised dataset is very low and IPO performs better than other methods under a low proportion of NPs. These empirical results demonstrate that our method has good compatibility with various offline alignment methods.

**Compatibility with robust alignment methods.**    Table 6 shows the average reward accuracy and improvements of robust alignment methods compared to those combined with PerpCorrect. Our method, PerpCorrect, can significantly enhance the performance of cDPO [21], and provide a modest improvement for rDPO [8] under almost all proportion of NPs. The empirical results show that our method has good compatibility with robust alignment methods.

## 5    Conclusions

This paper proposes a method called perplexity-aware correction (PerpCorrect), as an effective approach for robust alignment with noisy preferences (NPs). PerpCorrect utilizes a surrogate LLM to calculate a novel metric, PPLDiff, and further detects and corrects NPs from clean preferences (CPs) based on it. PerpCorrect consists of three steps: (1) PerpCorrect aligns a surrogate LLM using the clean validation dataset, enabling PPLDiff to distinguish between CPs and NPs. (2) PerpCorrect enhances the discrimination power of PPLDiff by aligning the surrogate LLM with more reliable training data. (3) PerpCorrect detects and corrects NPs from CPs based on a calculated threshold and obtains a denoised training dataset. The paper further proposes a robust alignment pipeline, consisting of three stages SFT, PerpCorrect, and alignment, to achieve robust alignment with NPs. The experimental results validate that PerpCorrect achieves state-of-the-art alignment performance and has good compatibility with other online, offline, and robust alignment methods. Therefore, PerpCorrect can be an effective method to mitigate the impact of NPs and can be used for robust alignment. Future research directions include: (1) Improving the time efficiency of PerpCorrect and (2) Reducing the amount of clean validation data required to achieve the same alignment performance.

## Limitations

We discuss some limitations of this work to stimulate further research in this direction. Our limitations mainly stem from two aspects: time efficiency issues caused by multiple calculations of PPLDiff and repeated training of a surrogate LLM, and the need for a validation dataset.

**Time efficiency.**    Iteratively calculating the PPLDiff value for each data point and aligning a surrogate LLM is time-consuming. Selecting reliably training data and denoising the training dataset requires that the PPLDiff value be calculated for each data point during each epoch, which may cause unnecessary calculations for CPs and NPs that can already be clearly distinguished. Besides, aligning a surrogate LLM with same size as the LLM for alignment multiple times is time-consuming. The detailed discussion is in the Appendix B.

**Validation dataset.**    PerpCorrect requires a validation dataset for aligning a surrogate LLM. However, manually annotating a validation dataset is complex and labor-intensive in practice. As shown in Table 5, there is a significant disparity in alignment performance when comparing the use of 10 clean samples to 50 clean samples. Exploring how to use fewer clean samples or even no clean samples to achieve the same or better performance is a problem worth further investigation.

## Acknowledgements

This research is supported by the National Research Foundation, Singapore under its Strategic Capability Research Centres Funding Initiative, the funding BAS/1/1689-01-01, URF/1/4663-01-01, REI/1/5232-01-01, REI/1/5332-01-01, and URF/1/5508-01-01 from KAUST, and funding from KAUST - Center of Excellence for Generative AI, under award number 5940. Any opinions, findings and conclusions or recommendations expressed in this material are those of the author(s) and do not reflect the views of National Research Foundation, Singapore.

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

# A   Broader Impacts

Our proposed PerpCorrect and robust alignment pipeline offers a solution for achieving state-of-the-art performance in robust alignment under noisy preferences. PerpCorrect is designed to effectively reduce malicious noise in the dataset and mitigate biases introduced by human annotators, ensuring that the trained language model (LLM) is accurately aligned with true human preferences.

Moreover, we recognize a potential risk: if malicious users exploit our method for reverse training, they might compromise the security mechanisms of existing open-source LLMs. Existing research has demonstrated the possibility of reverse training [36].

# B   Time Efficiency Analysis

The additional computational overhead is primarily attributed to PerpCorrect (Section 3.1). Table 7 presents both theoretical and empirical runtime comparisons, where $X$ represents the theoretical time required for Alignment (Section 3.2) or other baselines.

In theory, during the PerpCorrect process, we need to calculate PPLDiff and train the surrogate model in each epoch. The computation time introduced by PerpCorrect is approximately $\frac{T}{3}$ that of the Alignment or other baselines.

The calculation of PPLDiff in each epoch requires only $\frac{1}{3}$ of the time needed for robust alignment. The primary computational load in robust alignment arises from the complexity of forwarding and back-propagation, while the complexities of gradient updates and parameter updates are relatively low. Additionally, back-propagation takes twice as long as forwarding. In addition, the calculation of PPLDiff only requires forwarding.

For surrogate model training, PerpCorrect utilized data points that represented $t \times \alpha$ of the total dataset during epoch $t$. Since both $t$ and $\alpha$ are small, the time required for surrogate model training can be approximately ignored.

In practice, Our entire robust alignment pipeline ($\sim$24 hours) takes only twice as long as the baseline ($\sim$12 hours). We set $T = 5$ and $\alpha = 2$, and used the AdamW optimizer. The practical efficiency of the PerpCorrect is due to the use of fp32 precision by the AdamW optimizer, which increases the GPU's calculation time during the robust alignment process.

Table 7: Comparison of theoretical and practical running times for PerpCorrect and baselines.

| Stage | Theoretical Running Time | Practical Running Time |
|---|---|---|
| PerpCorrect | $\frac{T}{3} \times X$ | $\sim$12 hours |
| Alignment (or baselines) | $X$ | $\sim$12 hours |
| Total | $(1 + \frac{T}{3})X$ | $\sim$24 hours |

# C   Implementation details

## C.1   Training details and compute resources.

We utilized the Qlora method [11] for fine-tuning the LLMs, executed on RTX 4090 GPUs with 24 GB of memory. Hyperparameters were set as follows: lora_rank = 32, lora_dropout = 0.1, and lora_alpha = 16. For SFT, we use the alpaca dataset [30] and set learning_rate = $2e - 4$ and batch_size = 20. For our PerpCorrect stage II, we set $\beta = 0.1$, learning_rate = $1e - 3$, batch_size = 4, $T = 5$, and $\alpha = 0.02$. For our PerpCorrect stage III and all other alignment methods, we set $\beta = 0.1$, learning_rate = $3e - 4$, and batch_size = 20. Other details not mentioned, we follow the default setting in TRL library. Each experiment, involving a specific method and proportion of NPs, could be completed using a single RTX 4090 GPU within 24 hours on the Golden HH dataset and within 72 hours on the OASST1 dataset.

## C.2 Description and Processing Details of the Datasets

**OpenAssistant Conversations Dataset (OASST1).** The original OASST1 dataset [17] is an assistant-style conversation corpus generated and annotated by humans. It consists of over 10,000 fully annotated conversations in 35 different languages. Sileo [28] converted these conversations into a preference dataset comprising 17,966 training samples and 952 testing samples. After filtering out conversations with one or fewer letters, we obtained a preference dataset with 17,939 training samples and 951 testing samples.

**Golden HH.** The Golden HH dataset [7] is a variant of the Anthropic Helpful and Harmless (HH) dataset [4]. It originally contains 42,537 training samples and 2,312 testing samples as part of a preference dataset. Each sample has two keys: one representing the prompt $x$ and the chosen response $y_w$, and the other representing the prompt $x$ and the rejected response $y_l$. We first converted the dataset into a triple form: prompt $x$, chosen response $y_w$, and rejected response $y_l$, retaining only one-turn conversation data. After filtering out samples with one or fewer letters, we obtained a preference dataset with 12,066 training samples and 654 testing samples.

## C.3 Detailed Robust Alignment via Perplexity-aware Correction

**Supervised Fine-Tuning (SFT).** The objective of Supervised Fine-Tuning (SFT) is to enhance the performance of a pre-trained large language model (LLM) by refining its abilities for specific tasks. As demonstrated by prior work [9, 25, 24], this can be achieved by utilizing supervised fine-tuning with a specialized dataset tailored to the target task. The SFT dataset is annotated with labels, providing examples that are directly relevant to the task. Specifically, for each data point $(x, y)$ in the SFT dataset, $x$ represents the prompt given to the LLM, and $y$ represents the expected response that the model should generate based on the prompt $x$. The process involves fine-tuning the LLM by maximizing the log-likelihood of the correct responses $y$ given the prompts $x$. Through this method, the model learns to produce more accurate and task-specific outputs, thereby significantly improving its performance on the given task.

**Perplexity-aware Correction (PerpCorrect).** We demonstrate the entire PerpCorrect algorithm in Algorithm 2.

**Alignment.** We can achieve alignment using the denoised training dataset $\tilde{\mathcal{D}}_{\text{denoised}}$ with an estimated proportion of NPs $\varepsilon'_{\text{denoised}}$. For offline alignment methods such as DPO, SLiC, and IPO, we can directly optimize the LLM using the denoised training dataset $\tilde{\mathcal{D}}_{\text{denoised}}$ based on the loss functions defined in Eqs. 6–8. For loss-based robust alignment methods, including cDPO and rDPO, we set $\varepsilon' = \varepsilon'_{\text{denoised}}$ and then optimize the LLM using the denoised training dataset $\tilde{\mathcal{D}}_{\text{denoised}}$ according to the loss functions mentioned in Eqs. 9 and 10. For the online alignment method RLHF (PPO), we first train a reward model using the denoised training dataset $\tilde{\mathcal{D}}_{\text{denoised}}$ based on the loss function described in Eq. 3. Subsequently, we further optimize the LLM using PPO according to the objective function detailed in Eq. 4.

# D Extended Experimental Results

## D.1 Experiment Statistical Significance

Tables 8–12 demonstrates the standard deviation of the reward accuracy reported in Tables 1–4 and 6.

## D.2 Average PPLDiff Values of Data from Different Datasets Calculated by Unaligned LLMs

We randomly selected 10,000 data points from each dataset and calculated PPLDiff using different LLMs. The datasets and LLMs are downloaded from the Huggingface website. The average PPLDiff values are reported in the Table 13.

Table 8: Standard deviation of reward accuracy for DPO series alignment methods using Llama2-7B on the Golden HH dataset. The average reward accuracy is reported in Table 1.

| Method | Proportion of noisy preferences (%) | | | |
|---|---|---|---|---|
| | 10 | 20 | 30 | 40 |
| vanilla DPO | 0.81% | 0.40% | 2.52% | 2.60% |
| cDPO | 1.15% | 0.81% | 1.76% | 1.64% |
| rDPO | 0.26% | 1.53% | 0.95% | 1.92% |
| PerpCorrect-DPO | 0.63% | 0.87% | 1.73% | 0.63% |

Table 9: Standard deviation of reward accuracy for PPO series alignment methods using Llama2-7B on the Golden HH dataset. The average reward accuracy is reported in Table 2.

| Method | Proportion of noisy preferences (%) | | | |
|---|---|---|---|---|
| | 10 | 20 | 30 | 40 |
| vanilla PPO | 0.15% | 1.30% | 4.05% | 0.77% |
| cPPO | 0.15% | 1.53% | 4.61% | 5.89% |
| rPPO | 0.62% | 1.38% | 1.55% | 5.29% |
| PerpCorrect-PPO | 0.35% | 1.15% | 1.34% | 1.57% |

Table 10: Standard deviation of reward accuracy for DPO series alignment methods using phi-2 on the Golden HH dataset. The average reward accuracy is reported in Table 3.

| Method | Proportion of noisy preferences (%) | | | |
|---|---|---|---|---|
| | 10 | 20 | 30 | 40 |
| vanilla DPO | 0.92% | 0.63% | 1.28% | 0.98% |
| cDPO | 0.35% | 0.54% | 0.49% | 1.84% |
| rDPO | 0.30% | 0.15% | 1.67% | 7.82% |
| PerpCorrect-DPO | 0.81% | 1.27% | 0.63% | 1.07% |

Table 11: Standard deviation of reward accuracy for DPO series alignment methods using phi-2 on the OASST1 dataset. The average reward accuracy is reported in Table 4.

| Method | Proportion of noisy preferences (%) | | | |
|---|---|---|---|---|
| | 10 | 20 | 30 | 40 |
| vanilla PPO | 0.70% | 1.30% | 2.46% | 1.64% |
| cPPO | 0.46% | 0.87% | 0.56% | 0.91% |
| rPPO | 0.96% | 2.29% | 1.24% | 6.61% |
| PerpCorrect-PPO | 0.35% | 1.89% | 1.35% | 1.06% |

Table 12: Standard deviation of reward accuracy and improvements of the offline and robust alignment methods, as well as those combined with PerpCorrect, using Llama2-7B on the Golden HH dataset. The average reward accuracy is reported in Table 6.

| Method | Proportion of noisy preferences (%) | | | |
|---|---|---|---|---|
| | 10 | 20 | 30 | 40 |
| DPO | 0.81% | 0.40% | 1.28% | 0.98% |
| PerpCorrect-DPO | 0.63% | 0.87% | 1.73% | 0.63% |
| $\Delta$ | 0.98% | 0.89% | 0.97% | 1.98% |
| SLiC | 1.91% | 1.33% | 6.77% | 8.02% |
| PerpCorrect-SLiC | 1.76% | 1.45% | 1.16% | 1.50% |
| $\Delta$ | 0.63% | 0.38% | 5.62% | 6.52% |
| IPO | 0.84% | 0.63% | 3.48% | 0.32% |
| PerpCorrect-IPO | 0.32% | 1.68% | 0.23% | 1.25% |
| $\Delta$ | 0.98% | 2.29% | 3.26% | 1.10% |
| cDPO | 1.15% | 0.81% | 1.76% | 1.64% |
| PerpCorrect-cDPO | 0.69% | 0.75% | 0.81% | 0.72% |
| $\Delta$ | 0.47% | 1.31% | 1.85% | 2.36% |
| rDPO | 0.26% | 1.53% | 0.95% | 1.92% |
| PerpCorrect-rDPO | 0.63% | 0.23% | 1.84% | 1.19% |
| $\Delta$ | 0.84% | 1.48% | 2.19% | 2.31% |

Table 13: Average PPLDiff values of randomly selected data points across datasets calculated by different LLMs. "Avg." refers to the average PPLDiff value over all the datasets.

| Model | Dataset | | | | Avg. |
|---|---|---|---|---|---|
| | HH-RLHF [4] | SafeRLHF [10] | SHP [12] | WebGPT [22] | |
| Qwen2-1.5B [35] | -0.140 | -0.002 | 0.040 | -0.018 | -0.030 |
| Qwen2-1.5B-Instruct [35] | -0.149 | -0.009 | 0.046 | -0.021 | -0.033 |
| Yi-1.5-6B [1] | -0.158 | -0.103 | 0.105 | -0.036 | -0.048 |
| Yi-1.5-6B-Chat [1] | -0.159 | -0.054 | 0.069 | -0.040 | -0.046 |
| gemma-2-2b [31] | -0.140 | -0.051 | 0.001 | -0.024 | -0.053 |
| gemma-2-2b-it [31] | -0.163 | -0.053 | 0.063 | -0.028 | -0.045 |
| falcon-7b [2] | -0.113 | -0.001 | 0.037 | -0.016 | -0.023 |
| falcon-7b-instruct [2] | -0.121 | 0.021 | 0.039 | -0.017 | -0.019 |
| Mistral-7B-v0.3 [16] | -0.133 | -0.045 | 0.048 | -0.026 | -0.039 |
| Mistral-7B-Instruct-v0.3 [16] | -0.201 | -0.058 | 0.065 | -0.038 | -0.058 |
| glm-4-9b [14] | -0.134 | -0.019 | 0.045 | -0.027 | -0.034 |
| glm-4-9b-chat-1m [14] | -0.135 | -0.019 | 0.049 | -0.028 | -0.033 |
| Llama-2-7b-hf [33] | -0.133 | -0.052 | 0.051 | -0.028 | -0.040 |
| Llama-2-7b-chat-hf [33] | -0.139 | -0.041 | 0.063 | -0.032 | -0.037 |

**Algorithm 2** Perplexity-aware Correction (PerpCorrect)

1: **Input:** Noisy training dataset $\tilde{\mathcal{D}}$, clean validation dataset $\mathcal{D}_{\text{val}}$, LLM $\pi_\theta$ parameterized by $\theta$
2: **Output:** Denoised training dataset $\tilde{\mathcal{D}}_{\text{denoised}}$ and estimated proportion of NPs $\varepsilon'_{\text{denoised}}$
3: $\pi_{\theta'} \leftarrow \pi_\theta, \mathcal{D}'_0 \leftarrow \emptyset, \varepsilon'_{\text{denoised}} \leftarrow 1, \tilde{\mathcal{D}}_{\text{denoised}} \leftarrow \tilde{\mathcal{D}}$,
4: **for** epoch $t = 0, \ldots, T$ **do**
5:     // Aligning the surrogate LLM
6:     $\pi_{\theta'} \leftarrow$ Alignment $(\pi_{\theta'}, \mathcal{D}'_t \cup \mathcal{D}_{\text{val}})$
7:     // Calculating the PPLDiff values for each data point
8:     $\Omega \leftarrow \emptyset$
9:     **for** $(\tilde{x}, \tilde{y}_w, \tilde{y}_l) \in \tilde{\mathcal{D}}$ **do**
10:       $z \leftarrow \log \text{PPL}(x + \tilde{y}_w; \theta') - \log \text{PPL}(x + \tilde{y}_l; \theta')$
11:       $\Omega \leftarrow \Omega \cup \{(\tilde{x}, \tilde{y}_w, \tilde{y}_l, z)\}$
12:     **end for**
13:     // Fitting PPLDiff density of noisy training dataset
14:     $\bar{\varepsilon}, \bar{\mu}, \bar{\sigma} \leftarrow$ Fitted parameters using Levenberg-Marquard algorithm with $\Omega$
15:     // Estimating NPs proportion of the denoised training dataset
16:     $\varepsilon'_{PC} \leftarrow$ Estimated proportion of NPs using the Eq.15 based on $\bar{\varepsilon}, \bar{\mu}, \bar{\sigma}$
17:     // Keeping denoised training dataset with the smallest $\varepsilon'_{\text{denoised}}$
18:     **if** $\varepsilon'_{PC} < \varepsilon'_{\text{denoised}}$ **then**
19:       $\varepsilon'_{\text{denoised}} \leftarrow \varepsilon'_{PC}$
20:       // Calculating the Threshold $\tau$
21:       $\tau \leftarrow$ x-coordinate of the intersection of the two normal distributions($\bar{\varepsilon}, \bar{\mu}, \bar{\sigma}$)
22:       // Distinguishing CPs and NPs based on the threshold $\tau$ and correcting NPs
23:       $\tilde{\mathcal{D}}_{\text{CPs}} \leftarrow \emptyset, \tilde{\mathcal{D}}_{\text{NPs}} \leftarrow \emptyset$
24:       **for** $(\tilde{x}, \tilde{y}_w, \tilde{y}_l, z) \in \Omega$ **do**
25:         **if** $z > \tau$ **then**
26:           $\tilde{\mathcal{D}}_{\text{CPs}} \leftarrow \tilde{\mathcal{D}}_{\text{CPs}} \cup \{(\tilde{x}, \tilde{y}_w, \tilde{y}_l)\}$
27:         **else**
28:           $\tilde{\mathcal{D}}_{\text{NPs}} \leftarrow \tilde{\mathcal{D}}_{\text{NPs}} \cup \{(\tilde{x}, \tilde{y}_l, \tilde{y}_w)\}$
29:         **end if**
30:       **end for**
31:       $\tilde{\mathcal{D}}_{\text{donised}} \leftarrow \tilde{\mathcal{D}}_{\text{CPs}} \cup \tilde{\mathcal{D}}_{\text{NPs}}$
32:     **end if**
33:     $\mathcal{D}_{\text{Clean}} \leftarrow \emptyset, \mathcal{D}_{\text{Noisy}} \leftarrow \emptyset$
34:     // Calculating the left bound $\tau_l$ and the right bound $\tau_r$
35:     $\tau_l \leftarrow (t-1) \cdot \alpha \cdot (1 - \bar{\varepsilon}) \cdot |\tilde{\mathcal{D}}|$-th smallest PPLDiff value in $\Omega$
36:     $\tau_r \leftarrow (t-1) \cdot \alpha \cdot \bar{\varepsilon} \cdot |\tilde{\mathcal{D}}|$-th largest PPLDiff value in $\Omega$
37:     // Finding extra reliable training data
38:     **for** $(\tilde{x}, \tilde{y}_w, \tilde{y}_l, z) \in \Omega$ **do**
39:       **if** $z < \tau_l$ **then**
40:         $\mathcal{D}_{\text{Clean}} \leftarrow \mathcal{D}_{\text{Clean}} \cup \{(\tilde{x}, \tilde{y}_w, \tilde{y}_l)\}$
41:       **end if**
42:       **if** $z > \tau_r$ **then**
43:         $\mathcal{D}_{\text{Noisy}} \leftarrow \mathcal{D}_{\text{Noisy}} \cup \{(\tilde{x}, \tilde{y}_l, \tilde{y}_w)\}$
44:       **end if**
45:     **end for**
46:     $\mathcal{D}'_{t+1} \leftarrow \mathcal{D}_{\text{Clean}} \cup \mathcal{D}_{\text{Noisy}}$
47: **end for**

