# OpenReview forum: "Perplexity-aware Correction for Robust Alignment with Noisy Preferences"
_NeurIPS.cc/2024/Conference — NeurIPS 2024 poster_

### Official Review · Reviewer_J5j5 · 2024-07-10

**Soundness:** 3
**Presentation:** 3
**Contribution:** 3
**Rating:** 7
**Confidence:** 4

**Summary:**

The paper proposes Perplexity-Aware Correction (PerpCorrect) to improve the alignment of large language models (LLMs) by detecting and correcting noisy preferences (NPs). PerpCorrect identifies NPs using the differences in perplexity (PPLDiff) between chosen and rejected responses. The method involves aligning a surrogate LLM with clean validation data, refining it with reliably clean and noisy data, and correcting NPs based on PPLDiff. Experiments demonstrate that PerpCorrect enhances alignment performance, is practical with a small amount of validation data, and is compatible with various alignment techniques.

**Strengths:**

1. This paper innovatively addresses the impact of noisy preferences on alignment from the perspective of correcting noisy data.

2. The experiments are comprehensive, which cover existing online, offline, and robust alignment methods, such as PPO, DPO, cDPO, and rDPO. The results show that PerpCorrect achieves the state-of-the-art robust alignment preference evaluated with different LLMs and datasets. Besides, PerpCorrect can be further combined with the other loss-based robust alignment methods.

**Weaknesses:**

1. The standard deviation of reward accuracy is not reported in Tables 3, 4, and 6.

**Questions:**

I have two minor questions about the experiments:

1. As shown in Tables 1 and 3, the reward accuracy of other DPO family alignment methods evaluated with Llama2-7B is better than that evaluated with phi-2 (2B). However, PrepCorrect performs better when evaluated on phi-2 (2B). Could you give some explanation for this phenomenon?

2. As shown in Table 6, PerpCorrect shows significant improvement when combined with other methods, such as SLiC, IPO, and cDPO, but less improvement when combined with rDPO.  Could you give some analyses of this phenomenon?

**Limitations:**

The paper explains its limitations in the appendix, such as time efficiency and the requirement for a validation dataset.

---

> ### Author Rebuttal · Authors · 2024-08-06
>
> Many thanks for your comments! Please find our replies below.
>
> ``[Reply to W1] Please refer to our general author rebuttal [R3].  ``
>
> ``[Reply to Q1] We provide a discussion about this phenomenon as follows. ``
>
> We found that the two models have similar proportion of NPs in the denoised dataset using PerpCorrent. The table below reports the proportion of NPs before and after PerpCorrect with phi-2 and Llama2, respectively. These empirical results suggest that PerpCorrect (Stage II) is not the main reason why PerpCorrect-DPO performs better when evaluated on phi-2 (2B).
>
> | Before        | After (Llama2) | After (phi-2) |
> | ------------- | -------------- | ------------- |
> | 9.86% (~10%)  | 6.05%          | 6.41%         |
> | 19.93% (~20%) | 9.76%          | 7.33%         |
> | 29.94% (~30%) | 14.14%         | 14.5%         |
> | 40.06% (~40%) | 13.98%         | 16.06%        |
>
> As shown in Table 3, the reward accuracy of vanilla DPO on the phi-2 is better than that on the Llama2-7B. We conjecture that the phi-2 model has better robustness and that is why PerpCorrect performs better when evaluated on phi-2.
>
> ``[Reply to Q2] The superior performance of rDPO and the gap between the estimated and actural proportion of NPs are the two main reasons.``
>
> **The superior performance of rDPO:** Under different proportions of NPs (ranging from 10% to 40%), rDPO demonstrated superior performance, achieving greater than 90% reward accuracy. This performance surpasses that of other methods such as cDPO, SliC, and IPO, leaving less room for improvement.
>
> **The gap between the estimated and actural proportion of NPs:** One reason why PerpCorrect-rDPO is not as effective as rDPO for certain proportions of NPs (10% and 20%) is that there is a gap between the estimated and actural proportion of NPs in the denoised dataset based on Eq. 15. We present the estimated and actual proportions of NPs in the denoised dataset in the table below.
>
> | The Estimated Proportion of NPs | The Actual Proportion of NPs |
> | ------------------------------- | ---------------------------- |
> | 4.36%                           | 6.05%                        |
> | 8.34%                           | 9.76%                        |

---

> > ### Comment · Reviewer_J5j5 · 2024-08-13
> >
> > I appreciate the authors' response, which has solved my concerns.

---

### Official Review · Reviewer_LN38 · 2024-07-11

**Soundness:** 4
**Presentation:** 4
**Contribution:** 4
**Rating:** 6
**Confidence:** 5

**Summary:**

This paper presents a novel method called PerpCorrect for robust alignment in large language models, particularly in the presence of noisy preferences in training data. PerpCorrect addresses noisy preferences by evaluating the perplexity difference (PPLDiff) between selected and rejected responses. By leveraging PPLDiff to identify and rectify these noisy preferences, PerpCorrect generates a denoised training dataset. Experimental results demonstrate that PerpCorrect achieves state-of-the-art alignment performance with only a small amount of validation data.

**Strengths:**

1. The paper is well-written, offering a comprehensive explanation of the proposed method, including detailed descriptions, mathematical formulas, and pseudocode.
2. The proposed method can be integrated with both existing online and offline alignment methods, enhancing their robustness without requiring significant modifications to their core methodologies.
3. PerpCorrect demonstrates strong performance with only a modest number of clean validation data points, which is beneficial in scenarios where such data is scarce or expensive to obtain.
4. The proposed method has been empirically tested on several datasets and models, consistently showing superior performance over traditional methods in the presence of noisy data.

**Weaknesses:**

1. Tables 3 and 4 do not include the standard deviations for reward accuracy.
2. Despite requiring fewer validation data points, its performance and accuracy still heavily depend on the availability of a high-quality clean validation dataset to initially train the surrogate LLM. Therefore, it is necessary to discuss potential solutions to this limitation.
3. The process of calculating perplexity differences (PPLDiff) for each data point and aligning a surrogate LLM multiple times could be computationally intensive and time-consuming. Therefore, this additional cost needs to be discussed.

**Questions:**

1. Will the PPLDiff calculate on the new dataset, using models similar to those used with other data (e.g., Llama-2-7b-chat), result in a normal distribution with a mean of 0?

2. What are the reasons that make PrepCorrect-IPO perform better than PrepCorrect-cDPO and PrepCorrect-rDPO? Please analyze this phenomenon.

**Limitations:**

As discussed in the paper.

---

> ### Author Rebuttal · Authors · 2024-08-06
>
> Many thanks for your comments! Please find our replies below.
>
> ``[Reply to W1] Please refer to our general author rebuttal [R3].``
>
> ``[Reply to W2] Please refer to our general author rebuttal [R2].``
>
> ``[Reply to W3] Please refer to our general author rebuttal [R1].``
>
> ``[Reply to Q1] We validated this assumption by conducting experiments on other datasets using various models. ``
>
> We randomly selected 10,000 data points from each dataset and calculated PPLDiff using different models.  The datasets and LLMs are downloaded from the Huggingface website.
>
> The following table report the mean of PPLDiff values.
>
> |          Model           | HH-RLHF [1] | SafeRLHF [2] | SHP [3] | WebGPT [4] |  Avg.  |
> | :----------------------: | :---------: | :----------: | :-----: | :--------: | :----: |
> |        Qwen2-1.5B        |   -0.140    |    -0.002    |  0.040  |   -0.018   | -0.030 |
> |   Qwen2-1.5B-Instruct    |   -0.149    |    -0.009    |  0.046  |   -0.021   | -0.033 |
> |        Yi-1.5-6B         |   -0.158    |    -0.103    |  0.105  |   -0.036   | -0.048 |
> |      Yi-1.5-6B-Chat      |   -0.159    |    -0.054    |  0.069  |   -0.040   | -0.046 |
> |        gemma-2-2b        |   -0.140    |    -0.051    |  0.001  |   -0.024   | -0.053 |
> |      gemma-2-2b-it       |   -0.163    |    -0.053    |  0.063  |   -0.028   | -0.045 |
> |        falcon-7b         |   -0.113    |    -0.001    |  0.037  |   -0.016   | -0.023 |
> |    falcon-7b-instruct    |   -0.121    |    0.021     |  0.039  |   -0.017   | -0.019 |
> |     Mistral-7B-v0.3      |   -0.133    |    -0.045    |  0.048  |   -0.026   | -0.039 |
> | Mistral-7B-Instruct-v0.3 |   -0.201    |    -0.058    |  0.065  |   -0.038   | -0.058 |
> |         glm-4-9b         |   -0.134    |    -0.019    |  0.045  |   -0.027   | -0.034 |
> |     glm-4-9b-chat-1m     |   -0.135    |    -0.019    |  0.049  |   -0.028   | -0.033 |
> |      Llama-2-7b-hf       |   -0.133    |    -0.052    |  0.051  |   -0.028   | -0.040 |
> |    Llama-2-7b-chat-hf    |   -0.139    |    -0.041    |  0.063  |   -0.032   | -0.037 |
>
> The following table report the standard deviation of PPLDiff values.
>
> |          Model           | HH-RLHF [1] | SafeRLHF [2] | SHP [3] | WebGPT [4] | Avg.  |
> | :----------------------: | :---------: | :----------: | :-----: | :--------: | :---: |
> |        Qwen2-1.5B        |    0.764    |    0.475     |  0.367  |   0.361    | 0.492 |
> |   Qwen2-1.5B-Instruct    |    0.798    |    0.491     |  0.385  |   0.369    | 0.511 |
> |        Yi-1.5-6B         |    1.119    |    0.614     |  0.564  |   0.500    | 0.699 |
> |      Yi-1.5-6B-Chat      |    0.855    |    0.493     |  0.417  |   0.373    | 0.534 |
> |        gemma-2-2b        |    0.885    |    0.517     |  0.702  |   0.373    | 0.619 |
> |      gemma-2-2b-it       |    0.975    |    0.535     |  0.467  |   0.378    | 0.588 |
> |        falcon-7b         |    0.631    |    0.400     |  0.324  |   0.325    | 0.420 |
> |    falcon-7b-instruct    |    0.648    |    0.436     |  0.342  |   0.344    | 0.442 |
> |     Mistral-7B-v0.3      |    0.719    |    0.411     |  0.332  |   0.313    | 0.444 |
> | Mistral-7B-Instruct-v0.3 |    0.981    |    0.482     |  0.385  |   0.343    | 0.548 |
> |         glm-4-9b         |    0.709    |    0.454     |  0.357  |   0.350    | 0.467 |
> |     glm-4-9b-chat-1m     |    0.719    |    0.456     |  0.367  |   0.356    | 0.474 |
> |      Llama-2-7b-hf       |    0.670    |    0.390     |  0.322  |   0.309    | 0.423 |
> |    Llama-2-7b-chat-hf    |    0.709    |    0.411     |  0.360  |   0.339    | 0.455 |
>
> ``[Reply to Q2] We conjecture the main reason is that IPO performs better than cDPO and rDPO under a low proportion of NPs. ``
>
> The proportion of NPs in the dataset corrected by our method is very low (**~10%**). We provide the proportion of NPs before and after using our method on the Golden HH dataset with the Llama-2 model in the table below.
>
> | Before        | After  | $\Delta$ |
> | ------------- | ------ | -------- |
> | 9.86% (~10%)  | 6.05%  | 3.81%    |
> | 19.93% (~20%) | 9.76%  | 10.17%   |
> | 29.94% (~30%) | 14.14% | 15.80%   |
> | 40.06% (~40%) | 13.98% | 26.08%   |
>
> According to Table 6, the performance of IPO under low NPs (10%) is the main reason for the good performance of PerpCorrect-IPO. PerpCorrect-cDPO and PerpCorrect-rDPO are not as effective as PerpCorrect-IPO because both cDPO and rDPO are robust alignment methods that cannot fully utilize CPs.
>
> [1] Training a Helpful and Harmless Assistant with Reinforcement Learning from Human Feedback. ArXiv 2022
>
> [2] Safe RLHF: Safe Reinforcement Learning from Human Feedback. ICLR 2024
>
> [3] Understanding Dataset Difficulty with V-Usable Information. ICML 2022
>
> [4] WebGPT: Browser-assisted question-answering with human feedback. ArXiv 2021

---

> > ### Comment · Reviewer_LN38 · 2024-08-13
> >
> > Thank you for your reply, my concerns have been resolved.

---

### Official Review · Reviewer_JfpF · 2024-07-13

**Soundness:** 3
**Presentation:** 2
**Contribution:** 3
**Rating:** 5
**Confidence:** 2

**Summary:**

The paper proposed a novel method to mitigate the preference noise in alignment. The authors first provide insights into how the PPLDiff can recognize the noisy preferences and then use the PPLDiff to select and correct noise preferences. Extensive experiments demonstrate that the method can significantly improve the robustness under high noise ratios.

**Strengths:**

* The paper proposed a novel detect and correct paradigm for noise-robust alignment.
* Novel insights were revealed for the failure with the noise preferences.

**Weaknesses:**

* Some writing is not clear.
   - In Line 50, "However, Mitchell [15] and Chowdhury et al. [6] overlooked the essential, differences between noisy and clean preferences, which is critical for mitigating the issue of NPs." What is the overlooked difference here? The logic is not clear.

**Questions:**

* In Line 50, "However, Mitchell [15] and Chowdhury et al. [6] overlooked the essential, differences between noisy and clean preferences, which is critical for mitigating the issue of NPs." What is the overlooked difference here? The logic is not clear.

**Limitations:**

No obvious limitations.

---

> ### Author Rebuttal · Authors · 2024-08-06
>
> Many thanks for your comments! Please find our replies below.
>
> ``[Reply to Q1] The overlooked difference is that NPs have incorrect labels, which can be identified using PPLDiff.``
>
> Both cDPO (ref to Eq. 9) and rDPO (ref to Eq. 10) use a universal loss to treat CPs and NPs. They overlooked the difference between CPs and NPs, that is, NPs have incorrect labels, while CPs have correct labels. This results in the underutilization of CPs and the negative impact of NPs.
>
> In contrast, our approach, PerpCorrect, addresses this issue more effectively by using PPLDiff to identify and correct NPs from a data perspective rather than a loss perspective.

---

> > ### Comment · Reviewer_JfpF · 2024-08-13
> >
> > Thank you for the response which addressed my concerns.

---

### Official Review · Reviewer_JSSg · 2024-07-14

**Soundness:** 4
**Presentation:** 3
**Contribution:** 3
**Rating:** 7
**Confidence:** 4

**Summary:**

The paper introduces Perplexity-aware Correction (PerpCorrect), a method for robust alignment of large language models (LLMs) with noisy preferences (NPs). PerpCorrect detects and corrects NPs by analyzing the perplexity difference (PPLDiff) between chosen and rejected responses. The approach involves aligning a surrogate LLM with clean validation data, iteratively refining with reliable training data, and ultimately producing a denoised training dataset. Experiments demonstrate that PerpCorrect significantly improves alignment performance and is compatible with various alignment techniques.

**Strengths:**

1. The problem of aligning LLMs with human preferences while effectively handling noisy data is a significant challenge in the field of AI, and this paper provides a valuable contribution towards solving it.

2. This paper introduces PPLDiff, a novel metric for distinguishing between clean and noisy preferences, enhancing the precision of corrections. The proposed method effectively handles noisy preferences (NPs), improving the reliability of LLM alignment.

3. PerpCorrect obtains significant improvements compared to previous methods, requiring only a modest amount of clean validation data, making it practical for real-world applications.

**Weaknesses:**

1. Implementing PerpCorrect involves additional steps of calculating and analyzing perplexity differences (PPLDiff), which can add complexity to the alignment process. Can the authors discuss the complexity and additional computation time of the method?

2. The method requires a clean validation dataset to align the surrogate LLM initially. The need for manually annotated validation data can be labor-intensive and may not always be feasible in large-scale applications.

3. The method assumes that the perplexity of clean and noisy preferences will differ consistently, which may not always hold true across all datasets and model configurations.

**Questions:**

See Weaknesses.

**Limitations:**

Limitations have been discussed in Conclusion section.

---

> ### Author Rebuttal · Authors · 2024-08-06
>
> Many thanks for your comments! Please find our replies below.
>
> ``[Reply to W1] Please refer to our general author rebuttal [R1].``
>
> ``[Reply to W2] Please refer to our general author rebuttal [R2].``
>
> ``[Reply to W3] We provide a discussion of the assumption as follows.``
>
> **Technically**,  this assumption is inspired by the loss function of DPO (As shown in Eqs. 5-6). According to Rafailov et al. [1], the gradient can be written as:
> $$
> \nabla\_\theta\mathcal{L}\_{\mathrm{DPO}}(\pi\_\theta;\pi\_{\mathrm{ref}})=-\beta\mathbb{E}\_{(x,y\_w,y\_l)\sim\mathcal{D}}\bigg[\sigma(\beta\log\frac{\pi\_\theta(y\_l|x)}{\pi_\text{ref}(y\_l|x)}-\beta\log\frac{\pi\_\theta(y\_w|x)}{\pi\_\text{ref}(y\_w|x)})\bigg[\nabla_\theta\log\pi(y\_w| x)-\nabla_\theta\log\pi(y\_l| x)\bigg]\bigg],
> $$
> where $\nabla_\theta\log\pi(y_w| x)$ increases likelihood of $y_w$ and $\nabla_\theta\log\pi(y_l| x)$ decreases likelihood of $y_l$. Furthermore, for CPs where $(x,\tilde{y_w},\tilde{y_l}) = (x,y_w,y_l)$, the gradient directs the decrease of $\mathrm{PPL}([x;\tilde{y}\_{w}];\theta)$ and the increase of $\mathrm{PPL}([x;\tilde{y}\_{l}];\theta)$. This ultimately leads to the reduction of $\mathrm{PPLDiff}(x,\tilde{y}\_w,\tilde{y}\_l;\theta)$. Conversely, for NPs, the gradient leads to an opposite effect. As a result, the PPLDiff of CPs and NPs will consistently differ.
>
> **Empirically**, the results of our method, PerpCorrect, using different LLMs (including phi-2 and Llama-2) on various datasets (such as the Golden HH and OASST1 datasets) have verified our assumption.
>
> [1] Direct Preference Optimization: Your Language Model is Secretly a Reward Model. NeruIPS 2023

---

### Author Rebuttal · Authors · 2024-08-06

We thank all reviewers for their insightful comments and suggestions. Please find our replies below.

``[R1] We discuss the complexity and additional computation time of method as follows.``

The additional computation time is primarily due to PerpCorrect. Both the theoretical and practical times are reported in the table below (using XXX to represent the theoretical time required for robust alignment) and are followed by a detailed analysis.

| Stage                                             | Theoretical Time       | Practical Time |
| ------------------------------------------------- | ---------------------- | -------------- |
| PerpCorrect (Stage II)                            | $\frac{T}{3} \times X$ | ~12 hours      |
| Robust alignment (Stage III and baseline methods) | $X$                    | ~12 hours      |
| Total                                             | $(1+\frac T3) X$       | ~24 hours      |

**In theory**, during the PerpCorrect process, we need to calculate PPLDiff and train the surrogate model in each epoch. The computation time introduced by PerpCorrect (Stage II) is approximately $\frac{T}{3}$ that of the robust alignment (Stage III and baseline methods).

- The calculation of PPLDiff in each epoch requires only $\frac{1}{3}$ of the time needed for robust alignment (Stage III). The primary computational load in robust alignment arises from the complexity of forwarding and back-propagation, while the complexities of gradient updates and parameter updates are relatively low. Additionally, back-propagation takes twice as long as forwarding. In addition, the calculation of PPLDiff only requires forwarding.
- For surrogate model training, PerpCorrect utilized data points that represented $t\times \alpha$ of the total dataset during epoch $t$. Since both $\alpha$ and $t$ are small, the time required for surrogate model training can be approximately ignored.

**In practice**, Our entire robust alignment pipeline (**~24h**) takes only **twice** as long as the baseline (**~12h**). We set $T=5$ and $\alpha=2\%$, and used the AdamW optimizer. The practical efficiency of the PerpCorrect is due to the use of fp32 precision by the AdamW optimizer, which increases the GPU's calculation time during the robust alignment process.

``[R2] We provide a discussion about the need for clean datasets and potential solutions as follows. ``

We only need 50 clean data points, which constitutes **less than 0.5%** of the entire dataset. In practice, such small-scale annotation is feasible. Reviewing and labeling data are essential steps to enhance data quality. For instance, Scale AI utilizes a large workforce to review and ensure the high quality of their data [1].

To address the challenge of obtaining sufficient clean datasets, one potential approach is to employ the LLM-as-a-judge method, which allows models to self-annotate and thereby reduce reliance on manually curated clean data. This method has been discussed in recent works [2] [3] [4], and could serve as a promising solution to this limitation.

``[R3] We repeated the experiments three times with the same settings for Tables 3, 4, and 6, and the average reward accuracy and standard deviation are reported in the PDF file.``

[1] Scale AI, https://scale.com/docs

[2] Self-Rewarding Language Models, ICML 2024

[3] Constitutional AI: Harmlessness from AI Feedback, Arxiv 2024

[4] Trustllm: Trustworthiness in large language models, ICML 2024

---

### Decision · Program_Chairs · 2024-09-25

**Decision:**

Accept (poster)

**Comment:**

All four reviewers provided positive recommendations for this work and recognized that the problem of noisy preference is significant, the proposed method is novel, the experiments are comprehensive. The reviewers also pointed out that it can be better if the authors can discuss the limitation of requiring high-quality clean validation dataset, and report the standard deviation in the final version. As a result, I recommend this paper for acceptance.